# The Semen Microbiome and Semen Parameters in Healthy Stallions

**DOI:** 10.3390/ani12050534

**Published:** 2022-02-22

**Authors:** Carlota Quiñones-Pérez, Amparo Martínez, Isabel Ortiz, Francisco Crespo, José Luis Vega-Pla

**Affiliations:** 1Laboratorio de Investigación Aplicada, Cría Caballar de las Fuerzas Armadas, Carretera de Madrid Km 395A, 14014 Córdoba, Spain; jvegpla@oc.mde.es; 2Genetics Department, Edificio Gregor Mendel (C5), Campus de Rabanales, University of Córdoba, 14071 Córdoba, Spain; amparomartinezuco@gmail.com; 3Veterinary Reproduction Group, Department of Animal Medicine and Surgery, Campus Rabanales, University of Córdoba, 14071 Córdoba, Spain; isabel.ortiz.vet@gmail.com; 4Centro Militar de Cría Caballar de Ávila, Cría Caballar de las Fuerzas Armadas, Calle Arsenio Gutiérrez Palacios, 05005 Ávila, Spain; fcrecas@oc.mde.es

**Keywords:** microbiome, stallion, sperm quality, motility

## Abstract

**Simple Summary:**

Stallion infertility is a major cause of concern in the horse industry. Despite zootechnics advances, sub- or infertile animals appear in stud farms without a toxic, genetic, or nutritional reason. Recent research in human andrology has opened the door for a new, plausible factor that affects sperm quality: seminal microflora. In recent years, there has been an increasing amount of evidence regarding the relationship between different seminal flora compositions and male fertility. However, little has been studied in veterinary science, including horses. Therefore, the objective of this study was to examine associations with the presence of bacteria families in horse semen with five sperm quality parameters: concentration, total number of spermatozoa, total and progressive sperm motility, and DNA fragmentation. Our study detected a correlation between the presence of the Peptoniphilaceae family and higher total motility and the presence of Clostridiales Incertae Sedis XI and lower progressive motility. These changes in seminal flora may contribute to the idiopathically poorer sperm quality in certain animals. Although further mechanisms behind bacteria–spermatozoa interactions are unknown, these associations are already leading to a new therapeutic approach to infertility: the use of prebiotics, which has already yielded promising results in human andrology.

**Abstract:**

Despite the advances in reproductive technology, there is still a considerable number of low sperm quality cases in stallions. Recent studies in humans have detected several seminal microflora–spermatozoa associations behind some idiopathic infertility cases. However, no studies are available on horses, and there is limited information on the microflora present in stallion ejaculates. Accordingly, the objective of this study was to examine associations to the presence of bacteria families with five sperm quality parameters: concentration, total number of spermatozoa, total and progressive motility, and DNA fragmentation. Samples were cryopreserved after their extraction. High-speed homogenization using grinding media was performed for cell disruption. Family identification was performed via 16S rRNA sequencing. Bacterial families were only considered if the relative abundance was higher than 1%. Only two families appeared to have a correlation with two sperm quality parameters. Peptoniphilaceae correlated positively with total sperm motility, whereas Clostridiales Incertae Sedis XI correlated negatively with progressive motility. No significant differences were found for the rest of the parameters. In conclusion, the seminal microbiome may affect spermatozoa activity. Our findings are based on statistical associations; thus, further studies are needed to understand the internal interactions between seminal flora and cells.

## 1. Introduction

The success of the equine industry greatly depends on good reproductive outcomes. These outcomes depend on a variety of factors, such as sperm quality. There are objective parameters that assess sperm quality, such as concentration, total motility, or progressive motility [1,2,3]. Factors affecting these parameters have been subject to large-scale analysis in horse reproductive science [4].

In recent years, the microbiome has proven to have a great impact on the systems they dwell on [5,6,7,8,9]. Unfortunately, the male reproductive tract has not received sufficient attention [10,11]. In the human species, however, some authors have already pointed out the influence of bacteria on semen quality [12,13,14,15,16]. In fact, these studies have opened the door to a potential therapeutic tool in infertility cases, and some authors have already published some positive effects of prebiotics in improving sperm quality [17,18,19,20].

Unfortunately, in veterinary science, research focuses on animal experimentation, such as mice [17] or broilers [18]. Regarding stallions, papers usually focus on pathogenic bacteria [21] or on their effect on reproductive technologies [22,23]. Besides, these are usually culture-based studies, which may underestimate the presence of some difficult-to-culture bacteria [24].

To the best of our knowledge, there are no studies evaluating sperm quality and the seminal microbiome in this species. Therefore, the objective of this study was to assess the relationship between the presence of more abundant bacteria and five sperm quality parameters: concentration, total number of spermatozoa, total and progressive sperm motility, and DNA fragmentation.

## 2. Materials and Methods

### 2.1. Materials

#### 2.1.1. Sample Collection

All the experiments were performed in accordance with the Spanish law for animal welfare and experimentation (Decision 2012/707/UE and RD 53/2013). Animals belonged to the Equine Breeding Centre of the Spanish Army of Écija. Animals lived in accordance with the Spanish law for animal welfare (Law 32/07). Semen was opportunistically collected during daily work to avoid extra collections.

Samples were collected from 12 clinically stallions (7 Andalusians, 4 Arabs, and 1 Anglo-Arab) in Écija (Seville, Spain) during the breeding season (March–June). Stallions were collected a maximum of 3 times per week, with at least 48 h between collections. Age ranged from 6 to 23 years old, mean 13.3 ± 5.2 standard deviation (Table 1). Semen collection was performed using a phantom for stallion support, with a mare in estrus to stimulate sexual behavior. Semen was collected using a Missouri-type artificial vagina (Minitüb^®^, Tiefenbach, Germany) with an in-line filter. In order to prevent contamination, personnel wore gloves during the whole process of collection, preparation, and evaluation of ejaculates. An inner disposable sterile plastic liner was used for each animal. It was internally spread with a sterile, silicon-free commercial lubricant (Vet Gel, Kerbl^®,^ Buchbach, Germany). At the beginning of the breeding season, the penis and prepuce of the stallion were gently washed with warm water to remove smegma excess. No routinary penis preparation prior to collection was performed unless there was smegma accumulation. Animals were housed in individual boxes with straw bedding, fed under the same dietary conditions, and had the same exercise regime. Diet included alfalfa hay, commercial concentrate, and oats. 

Each sample was divided into two aliquots to evaluate: (i) Sperm quality: raw semen was extended with INRA96^®^ (IMV, L’Aigle, France) until reaching 25 × 10^6^ sperm/mL to assess sperm parameters (Table 1); and (ii) Microbiome: raw semen was frozen immediately after collection following the method described in [25], prior to analysis using next-generation sequencing as detailed below. 

#### 2.1.2. Control Sample

In order to evaluate the extraction and amplification quality, the pattern sample ZymoBIOMICS Microbial Community Standard^®^ (Zymo Research, Irvine, CA, USA) was included during DNA extraction.

### 2.2. Methods

#### 2.2.1. Sperm Parameters Evaluation

Sperm concentration was measured using a spectrophotometer (Spermacue^®^, Minitüb, Tiefenbach, Germany). The total number of spermatozoa was calculated by multiplying the concentration and volume. Then, semen was diluted until reaching an approximate concentration of 25 × 10^6^ sperm/mL in milk-based extender (INRA 96^®^, IMV Technologies, L’Aigle, France) and placed in a 37 °C water bath. The extender contains fractions of milk micellar proteins, penicillin, gentamicin, and amphotericin B. Extended semen was only used for sperm parameters evaluation.

Sperm motility was evaluated using computer-assisted sperm analysis (Sperm Class Analyzer^®^, SCA, Microptic SL, Barcelona, Spain) using a 37 °C heated plate and a phase-contrast microscope (Optiphot-2, Nikon^®^, Tokyo, Japan). Chamber slides were pre-heated at 37 °C and up with the extended samples. Total (TM, %) and progressive sperm motility (PM, %) were evaluated as described by [25]. The minimum number of cells per sample analyzed was 500.

Sperm DNA fragmentation was assessed with the Sperm Halomax kit^®^ (Halotech DNA^®^ SL, Madrid, Spain), as described in [26].

#### 2.2.2. DNA Extraction

DNA extraction was performed using a ZymoBIOMICS^®^ DNA Miniprep (Zymo Research^®^, CA, USA) commercial kit. Samples had been previously submitted to a combination of mechanic and enzymatic-digestion cell disruption, as described by Bag [27]. Briefly, 100 μL of the raw semen sample was broken down for 1 h with 10 mg/mL lysozyme, 4000 U/mL lysophosphatin, and 25,000 U/mL mutanolysin. Then, samples were mechanically disrupted by high-speed homogenization (5000 rpm for 5 min) in grinding media (0.1 and 0.5 mm-diameter ceramics beads). Then, DNA was extracted following the manufacturer’s instructions.

#### 2.2.3. Next-Generation Sequencing Analysis

Next-generation analysis was performed using Ion semiconductor sequencing following the protocol described by Quiñones [28].

Data analysis was performed in the Ion Reporter server system (https://ionreporter.thermofisher.com/ir/secure/home.html) (accessed on 15 October 2021). Hypervariable region V3 was chosen for bacterial identification, as it has been suggested to detect a wider range of bacterial species [29].

#### 2.2.4. Statistical Analysis

Bray-Curtis dissimilarity was calculated among animals (Table 2) using the following formula: $${\mathrm{BC}}_{i,j}=1-\frac{2\times B_{i,j}}{\;A_{i}+B_{j}}$$

BC_i,j_ = Bray-Curtis dissimilarity.

B_i,j_ = sum of the lesser count of common families in groups A and B.

A_i_ = total number of bacterial families in group A.

B_j_ = total number of bacterial families in group B.

Values range from 0 to 1. Values closer to 1.00 mean more dissimilarity between groups. Statistical analysis was performed using Microsoft Excel^®^ 2013.

## 3. Results

Quality control was performed by submitting the ZymoBIOMICS Microbial Community Standard^®^ to the same extraction and analysis process as the rest of the samples. The resulting composition showed minor variations compared to that provided by the manufacturer (Figure 1).

Then, samples were submitted to a sperm quality analysis. Concentration, number of spermatozoa, total and progressive sperm motility parameters, and sperm DNA fragmentation were included. Results are represented in Table 1.

Four common phyla were detected in samples and a total of 74 families. Phyla results are represented in Figure 2.

Then, Bray-Curtis dissimilarity was calculated between animals. Results are represented in Table 2.

## 4. Discussion

Our results show that there might be a correlation between some sperm quality parameters and the seminal flora composition of healthy, fertile stallions, in particular, Firmicutes phylum. Although there are some individual differences, the more abundant phyla are common in all animals. Starting with Firmicutes phylum, the literature contains divergent findings regarding the effect of bacterial families on sperm quality. Some authors have highlighted the positive correlation between specific Firmicutes families and good sperm quality. In this regard, *Lactobacillus* gender has been proven to have a protective effect on spermatozoa [13,30,31,32]. The mechanisms of protection are not fully understood, but they may be related to the antioxidant products exerted by lactobacilli in the extracellular environment [32]. Additionally, the positive effects of lactobacilli supplementation on sperm quality parameters have also been described in humans [19,20], mice [17], and broilers [18]. Stallion semen is not abundant in the *Lactobacillus* genus [10,11], but there are related bacterial families.

However, other authors have found some Firmicutes to have a detrimental effect on sperm parameters. In the literature, we found *Anaerococcus*, a Clostridiales genus, to have a detrimental effect on sperm quality [15,33]. The underlying mechanism needs to be further studied. Another detrimental family is Mycoplasmataceae (specifically, its *Ureaplama* genus) [24,34,35,36], whose pathogenic activity lies in acrosome damage [35]. This family was not detected in our samples.

A dominant family in fertile stallions is Porphyromonadaceae [11]. According to our results, this family is highly abundant, as it represents almost the whole Bacteroidetes phylum. This family seems to be a natural component in fertile males [15,34]; however, it is not as abundant as it is in horses. Regarding this family, it is necessary to highlight that it is difficult to find in culture-based references, as it is a laborious process to culture bacteria. The other highly dominant family in stallion semen is Corynebacteriaceae [10,11], which has regularly been found in fertile individuals [10,23,37,38,39,40]. However, some authors consider it has an opportunistic character [22,33,41]. It has been associated with a higher activity of caspases [22], which is usually linked to apoptosis [42]. Its predominance has also been linked to low motility [43]. 

Most infertility-related bacteria families belong to the Proteobacteria phylum, particularly to the Gammaproteobacteria class [13,24,35,44]. Enterobacteriaceae is included in this group, which has been found to alter spermatozoa motility [22,35] and the proportion of dead spermatozoa [22]; and Pseudomonadaceae [13,24,35,39,40], which may contain opportunistic pathogenic species [13,45]. In stallions, the presence of the Enterobacteriaceae family typically has a fecal origin [46] and worsens various seminal parameters [22,39,40]. Regarding Pseudomonadaceae, this family has been related to lower values of motility and integrity parameters [23,46,47], while other authors agree to consider it an opportunistic pathogen [40]. However, this family has been regularly found in the semen of fertile stallions [46,48]. The negative impact of these two families has also been found in boars [45]. In our case, Enterobacteriaceae only appeared in one horse (0.19%) and Pseudomonadaceae in just two (0.06% and 0.04%).

Bacteria in stallion semen have long been associated with a detrimental impact on fertility [23] as well as with a lower storage capacity [22]. Our study wanted to show that there may be certain bacterial families that harmoniously dwell in semen.

Regarding the strengths and limitations of our paper, next-generation sequencing is a better tool to characterize the seminal flora, as it overcomes laborious-to-culture bacteria [10,11,28,49]. However, the data process may be more complicated [50]. Finally, we mostly compared our findings with those of experiments carried out on humans, as animal references are extremely scarce. Further studies including a larger number of animals, including subfertile stallions with low sperm quality, are needed in order to find the possible relationship between the seminal microbiome and sperm quality.

## 5. Conclusions

In conclusion, four common bacterial phyla are present in all the stallions evaluated: Firmicutes, Bacteroidetes, Actinobacteria, and Proteobacteria. Although proportions vary among individuals, sperm quality values are similar. Further studies are needed to better understand the interactions between seminal flora and sperm quality.

## Figures and Tables

**Figure 1 animals-12-00534-f001:**
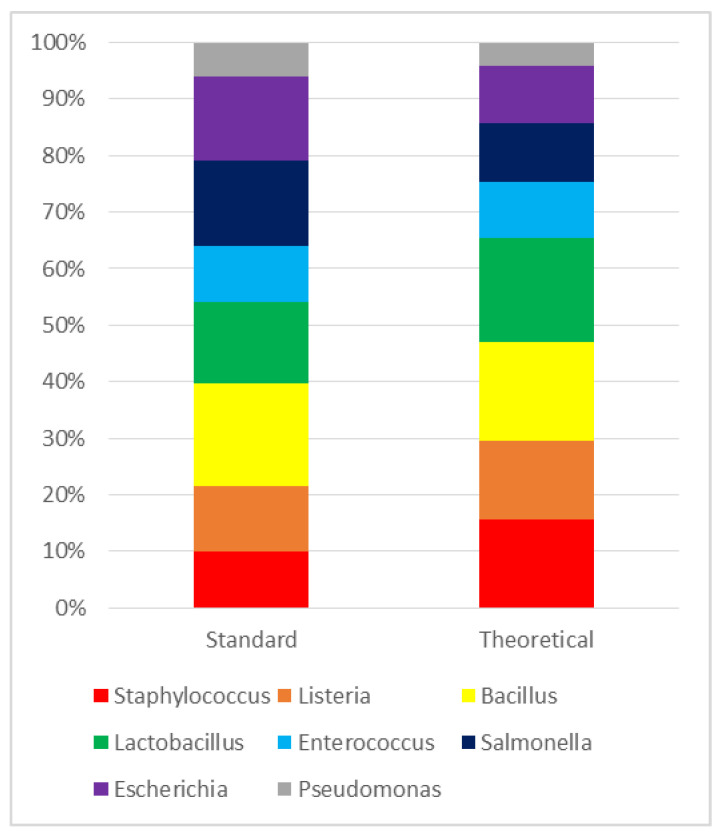
Bacterial composition of standard sample compared to its theoretical composition. Results are expressed as a percentage (%).

**Figure 2 animals-12-00534-f002:**
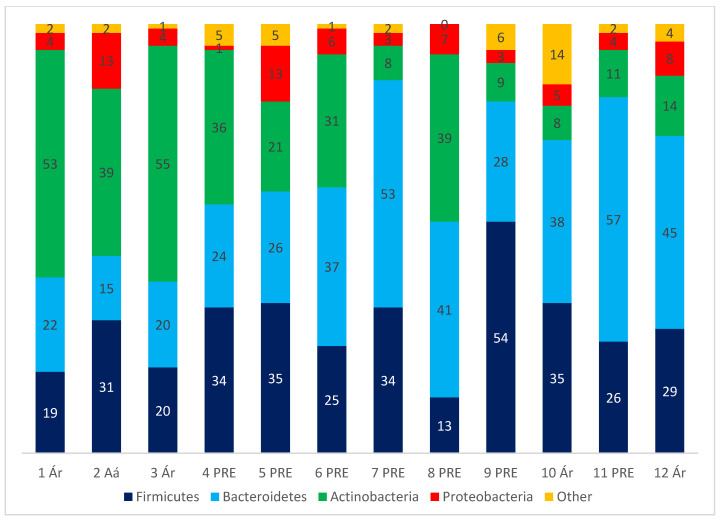
Graphical representation of bacteria the four more abundant phyla in samples. Numbers in X axis represent animals. PRE: Andalusian. Aa: Anglo-Arabian. Ar: Arabian. Numbers inside the bars represent the percentage of abundance of each phyla.

**Table 1 animals-12-00534-t001:** Sperm quality analysis: Numbers represent animals. PRE: Andalusian. Aa: Anglo-Arabian. Ar: Arabian. C: sperm concentration (millions of cells/mL). NSPZ: total number of sperm (millions). TM: total sperm motility (%). PM: progressive sperm motility (%). Frag: sperm DNA fragmentation (%).

	Breed	C	NSPZ	TM	PM	Frag
1	Ar	163	6520	80.0	38.0	6.0
2	Aa	79	6715	80.0	40.0	6.7
3	Ar	372	3348	91.0	42.0	4.3
4	PRE	232	9280	70.0	25.0	11.7
5	PRE	227	6810	75.0	25.0	8.3
6	PRE	374	9350	80.0	50.0	3.3
7	PRE	220	12,100	90.0	25.0	5.0
8	PRE	377	3770	77.0	43.0	8.0
9	PRE	392	7840	75.0	25.0	4.0
10	Ar	307	6140	68.0	36.0	7.7
11	PRE	230	8050	85.0	57.0	5.0
12	Ar	339	2712	94.0	38.0	3.0

**Table 2 animals-12-00534-t002:** Bray-Curtis dissimilarity. Values closer to 1.00 mean more dissimilarity between samples. Numbers represent animals. PRE: Andalusian. Aa: Anglo-Arabian. Ar: Arabian. Values range from 0 to 1. Values closer to 1.00 mean more dissimilarity between groups.

	Breed	1	2	3	4	5	6	7	8	9	10	11	12
1	Ar	-	-	-	-	-	-	-	-	-	-	-	-
2	Aa	0.21	-	-	-	-	-	-	-	-	-	-	-
3	Ar	0.02	0.21	-	-	-	-	-	-	-	-	-	-
4	PRE	0.20	0.15	0.22	-	-	-	-	-	-	-	-	-
5	PRE	0.32	0.18	0.34	0.15	-	-	-	-	-	-	-	-
6	PRE	0.23	0.22	0.24	0.18	0.21	-	-	-	-	-	-	-
7	PRE	0.46	0.41	0.48	0.31	0.27	0.26	-	-	-	-	-	-
8	PRE	0.22	0.26	0.24	0.26	0.44	0.13	0.35	-	-	-	-	-
9	PRE	0.45	0.40	0.47	0.27	0.22	0.35	0.25	0.46	-	-	-	-
10	Ar	0.45	0.39	0.47	0.36	0.21	0.24	0.10	0.33	0.20	-	-	-
11	PRE	0.42	0.42	0.44	0.36	0.31	0.22	0.04	0.31	0.32	0.22	-	-
12	Ar	0.39	0.32	0.37	0.28	0.19	0.17	0.13	0.25	0.27	0.16	0.12	-

## Data Availability

Data are contained within the article.

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
