# Peer review of "The Semen Microbiome and Semen Parameters in Healthy Stallions"

_animals, 2022, doi:10.3390/ani12050534_

Round 1

Reviewer 1 Report

The aim of the study was to associate the presence of more abundant bacteria families with five sperm quality parameters: volume, concentration, total number of spermatozoa, total motility, progressive motility and DNA fragmentation.

The topic is interesting, but the manuscript has a lot of methodological flaws, I think there are a few things that need to be corrected before final publication, as the lack of relevant information makes it difficult to repeat the experiment and understand its results correctly.

In the Materials and Methods section, the animals from which the biological material was collected should be characterized in detail. What was their age and breed (especially since the authors provide this information in the Results chapter)? There is no information about the period in which the research was carried out, how often the semen was collected from the animals, and whether only the first ejaculate was collected?

Line 77 please provide the exact procedure for freezing and thawing semen or refer to the published methodology.

Lines 86 and 88 are the same, just join them together to organize this paragraph better. It is also worth adding to what final concentration the semen was diluted. The other important and omitted issue is the diluent and its composition. Most of the commercially available extenders contain antimicrobial substances. So what was the composition of the diluent used in this study?

Lines 91-98 what was the concentration of sperm in the tested sample? Was semen further diluted for CASA analysis? If it was diluted with which diluent?

Lines 121-125 I did not find information on how many times the tests were repeated? This is a very important point.

Fig. 1 the graph is difficult to read, try another variant.

Lines 137-141 is not entirely clear why the authors divided into two groups, please add information about qualitative groups (like as you named in tables).  Besides, in my opinion, this paragraph should rather be presented in the M&M chapter.

 Tables 1 and 2 are for fresh or frozen semen? this should be noted. If these are only results after cryopreservation, it would also be useful to show the microbiological changes in fresh semen.

Author Response

Reviewer 1:

The aim of the study was to associate the presence of more abundant bacteria families with five sperm quality parameters: volume, concentration, total number of spermatozoa, total motility, progressive motility and DNA fragmentation.

The topic is interesting, but the manuscript has a lot of methodological flaws, I think there are a few things that need to be corrected before final publication, as the lack of relevant information makes it difficult to repeat the experiment and understand its results correctly.

We thank the reviewer for their work on this paper and highly appreciate the comments and suggestions, which significantly contributed to improving the quality of this manuscript. Please, find below a detailed response to each of the comments:

  1. In the Materials and Methods section, the animals from which the biological material was collected should be characterized in detail. What was their age and breed (especially since the authors provide this information in the Results chapter)? There is no information about the period in which the research was carried out. Semen was collected during breeding season (spring and summer), information included (line 78) how often the semen was collected from the animals, During breeding season, animals are submitted to a maximum of three collections per week, with a 48 hour interval between collections (information included in lines 79) and whether only the first ejaculate was collected? That is correct, the ejaculates used for the study belonged to the ejaculate of one of the days of semen demand.
  2. Line 77 please provide the exact procedure for freezing and thawing semen or refer to the published methodology. Thank you for the correction. Reference included (López-Fernández et al., 2007, line 114).

  1. Lines 86 and 88 are the same, just join them together to organize this paragraph better. Correction done (line 99) It is also worth adding to what final concentration the semen was diluted. Semen was diluted to a final concentration of 25 x 106 sperm/ml (line 100) in order to assess sperm quality parameters (semen is usually too concentrated to evaluate motility and DNA fragmentation). The other important and omitted issue is the diluent and its composition. Most of the commercially available extenders contain antimicrobial substances. So what was the composition of the diluent used in this study? Information about diluent composition included in lines 102-103. We would like to point out that diluent was used exclusively to evaluate the sperm quality parameters (volume, concentration, total number of spermatozoa, total motility, progressive motility and DNA fragmentation). For the microbiome analysis we used cryopreserved (line 89), raw semen, in order to avoid antibiotic interactions.

  1. Lines 91-98 what was the concentration of sperm in the tested sample? Was semen further diluted for CASA analysis? If it was diluted with which diluent? Diluted semen was used for assessing sperm quality parameters. We submit samples to a final concentration of 25 x 106 sperm/ml (line 100). The Information about its composition has been included in lines 102-103.

  1. Lines 121-125 I did not find information on how many times the tests were repeated? This is a very important point. We obtained just one ejaculate per animal, as seminal microbiome has proven to remain constant through time [1,2]. The statistic analysis was then performed between groups of similar sperm quality values. Table 1 changed to make grouping clearer.

  1. Fig. 1 the graph is difficult to read, try another variant. The graph has been changed. Please, let us know what else we can modify for better understanding of data.

  1. Lines 137-141 is not entirely clear why the authors divided into two groups, please add information about qualitative groups (like as you named in tables). Besides, in my opinion, this paragraph should rather be presented in the M&M chapter. Some cut-off values were chosen according to Love et al. [3]. Based on previous studies, our laboratory only ships semen doses with progressive motility > 30%. DNA fragmentation does not have a consensual cut-off value; but, we used previous studies as a reference [4,5]. We have not found a consensual cut-off value for volume; we inferred one considering sperm concentration and total number of spermatozoa. This paragraph and table have been moved to M&M section.

  1. Tables 1 and 2 are for fresh or frozen semen? this should be noted. If these are only results after cryopreservation, it would also be useful to show the microbiological changes in fresh semen. Quality parameters (Table 1) were measured in fresh semen. Seminal microflora was assessed in semen after cryopreservation. Although there are not many papers available, current knowledge states that microflora is stable if samples are frozen immediately after their obtaining [1,2,6–8]. Besides, it would have been impossible for us to include information about microflora in fresh semen, as we do not have the data.

References:

  1. Al-Kass, Z.; Eriksson, E.; Bagge, E.; Wallgren, M.; Morrell, J.M. Bacteria Detected in the Genital Tract, Semen or Pre-Ejaculatory Fluid of Swedish Stallions from 2007 to 2017. Acta Vet. Scand. 2019, 61, 25, doi:10.1186/s13028-019-0459-z.
  2. Al-Kass, Z.; Guo, Y.; Vinnere Pettersson, O.; Niazi, A.; Morrell, J.M. Metagenomic Analysis of Bacteria in Stallion Semen. Anim Reprod Sci 2020, 221, 106568, doi:10.1016/j.anireprosci.2020.106568.
  3. Love, C.C. Sperm Quality Assays: How Good Are They? The Horse Perspective. Anim Reprod Sci 2018, 194, 63–70, doi:doi: 10.1016/j.anireprosci.2018.04.077.
  4. Sergerie, M.; Laforest, G.; Bujan, L.; Bissonnette, F.; Bleau, G. Sperm DNA Fragmentation: Threshold Value in Male Fertility. Hum. Reprod. 2005, 20, 3446–3451, doi:10.1093/humrep/dei231.
  5. D’Occhio, M.J.; Hengstberger, K.J.; Johnston, S.D. Biology of Sperm Chromatin Structure and Relationship to Male Fertility and Embryonic Survival. Animal Reproduction Science 2007, 101, 1–17, doi:10.1016/j.anireprosci.2007.01.005.
  6. Hou, D.; Zhou, X.; Zhong, X.; Settles, M.L.; Herring, J.; Wang, L.; Abdo, Z.; Forney, L.J.; Xu, C. Microbiota of the Seminal Fluid from Healthy and Infertile Men. Fertil. Steril. 2013, 100, 1261–1269, doi:10.1016/j.fertnstert.2013.07.1991.
  7. Weng, S.-L.; Chiu, C.-M.; Lin, F.-M.; Huang, W.-C.; Liang, C.; Yang, T.; Yang, T.-L.; Liu, C.-Y.; Wu, W.-Y.; Chang, Y.-A.; et al. Bacterial Communities in Semen from Men of Infertile Couples: Metagenomic Sequencing Reveals Relationships of Seminal Microbiota to Semen Quality. PLoS ONE 2014, 9, e110152, doi:10.1371/journal.pone.0110152.
  8. Quiñones-Pérez, C.; Hidalgo, M.; Ortiz, I.; Crespo, F.; Vega-Pla, J.L. Characterization of the Seminal Bacterial Microbiome of Healthy, Fertile Stallions Using next-Generation Sequencing. Anim Reprod 2021, 18, e20200052, doi:10.1590/1984-3143-AR2020-0052.

Reviewer 2 Report

Effect of Seminal Microbiome on Sperm Quality Parameters in the Stallion

Authors: Carlota Quiñones-Pérez,, Amparo Martínez, Isabel Ortiz, Francisco Crespo and JoséLuis Vega-Pla

The stated objective was: to review the associations between the presence of bacteria and sperm quality parameters

Methods: the author report that they have semen collected from 21 stallions. The semen is collected in a plastic liner, filtered and frozen. The semen in the extended and evaluated for various quality parameters such as ejaculate volume, sperm total motility, progressive motility, and sperm concentration. Authors use Next Generation Sequencing to study the microbiome of stallion semen and look at associations with various semen parameters. They have divided the stallion semen into lower and higher quality samples.

The Main concern will this study is that it is really a descriptive study of the semen microbiome of healthy stallions. There is an attempt however to assign sperm parameters to semen quality and fertility without actually linking them the literature, or to the stallion’s per cycle fertility. Fertility is best assessed through an evaluation of per cycle fertility.

The second concern is that the collection frequency of the stallion has been shown to influence the ejaculatory volume, sperm concentration, and sperm motility parameters. Yet these parameters are used to divide the stallions into lower and higher quality ejaculates but the collection frequency is not discussed. The collection frequency could very clearly have an impact on the percentage of the different bacteria in the samples.

The Methodology is not crystal clear.  The authors use the terms extraction, and collection. Semen extraction could refer to the collection of the semen or the extraction of the RNA. This needs to be clarified. It is not clear if extended semen or raw semen is evaluated.

The range of total motility (68-94%) in the stallion population is not very large. The authors report that is associated with a Peptoniphilaceae and higher motility and lower motility with Clostridiales Incertae Sedis X. Sperm Morphology was not assessed which could account for the variability between stallions in terms of sperm motility parameters.

The cut-offs used were not evidence based to indicate semen quality. No references are provided.  

Recommend that this data is better presented as a descriptive study on the semen microbiome and that the fertility data is presented to demonstrate the stallion’s have normal semen parameters.

Specific Comments

Title: Suggestion The Semen Microbiome and Semen Parameters in Healthy Stallions.

Objective: to review the associations between the presence of bacteria and sperm quality parameters. The experimental design does not allow this to be determined.

Section 3 Results

Line 67-68 please provide more information on what the stallions were fed. Trace minerals? Grain?

Line 70 and 76 Re use of the extraction, do you mean semen collection? Please clarify

Line 72-73 In preparation for semen collection was the stallion’s penis prepared or washed in any way? This is normally a standard procedure prior to semen collection. Please describe. Was a phantom used for semen collection or was the stallion collected off of the mare?

Line 75 Were the stallion’s individually housed? As the results don’t agree with other authors is this an environmental element?

Line 77 It would be better to state that the raw semen was frozen for analysis.

Line 85 Is this a sterile tube?

Line 87 Sperm motility is influenced by extenders such as INRA 96. Was the raw semen evaluated for motility? The same is true for DNA fragmentation, was the raw semen or extended semen used for the DNA fragmentation analysis?

Line 114 pooled? Describe what and how many samples were pooled?

Line 135 Why was sperm morphology not evaluated?

Line 137-141 A per cycle pregnancy rate is the most important statistic to use regarding stallion fertility.  Is this information available? How many mares were bred?  If so how many progressively motile morphologically normal sperm were the mares bred with? This should be used to divide the stallions into high and low rather than the arbitrary cut offs.

These cut-off values should have a biologic basis. What literature supports these cut-offs if any? All of these parameters are influenced by the frequency of semen collection. This would influence the category within which the stallion’s sperm parameters are placed. How do you account for this?

Line 145  - 146 Table 1: there are no red or green numbers on the Table. Samples were collected from 21 stallions why are there 12 stallions in the Table?

Line 149 Table how many stallions per group? You state you collected 21 stallions

Line 164 How many stallions per group? This method of data plotting should be changed to stacked bar graphs

Bray-Curtis dissimilarity plots?

Line 172 there will be a correlation if you calculate it , significant or not, but that does not prove causality.

The discussion does not appear to cover possible environmental factors as a cause of the variability in microbiome.

Author Response

Reviewer 2:

The stated objective was: to review the associations between the presence of bacteria and sperm quality parameters

Methods: the author report that they have semen collected from 21 stallions. The semen is collected in a plastic liner, filtered and frozen. The semen in the extended and evaluated for various quality parameters such as ejaculate volume, sperm total motility, progressive motility, and sperm concentration. Authors use Next Generation Sequencing to study the microbiome of stallion semen and look at associations with various semen parameters. They have divided the stallion semen into lower and higher quality samples.

We thank the reviewer for their work on this paper and highly appreciate the comments and suggestions, which significantly contributed to improving the quality of this manuscript. Please, find below a detailed response to each of the comments:

  1. The Main concern will this study is that it is really a descriptive study of the semen microbiome of healthy stallions. There is an attempt however to assign sperm parameters to semen quality and fertility without actually linking them the literature, or to the stallion’s per cycle fertility. Fertility is best assessed through an evaluation of per cycle fertility. That is correct, unfortunately, fertility cannot be predicted using sperm quality parameters. We have changed the terminology referring to fertility by high/medium/low semen quality. We have also changed the title, objective and conclusion of the paper, as you suggested. However, we would like to maintain the statistical part. It may point to a correlation between certain bacteria and that specific sperm parameters, as it has already been described in men [1,2]. We have included the limitation of the statistical analysis in the limitations part of the discussion (line 236-237).

  1. The second concern is that the collection frequency of the stallion has been shown to influence the ejaculatory volume, sperm concentration, and sperm motility parameters. Yet these parameters are used to divide the stallions into lower and higher quality ejaculates but the collection frequency is not discussed. The collection frequency could very clearly have an impact on the percentage of the different bacteria in the samples. Thank you for your comment. Collection frequency has an effect on sperm quality. Regarding bacteria composition, this effect is much less studied. However, some authors agree that it is less affected by continuous seminal collection [1,2] than regular quality parameters . To make results comparable, animals were similarly submitted to a collection regime of a maximum of three collections per week, with 48 hours between extractions.

  1. The Methodology is not crystal clear. The authors use the terms extraction, and collection. Semen extraction could refer to the collection of the semen or the extraction of the RNA. This needs to be clarified. This section has been rewritten for better understanding (lines 77-91) It is not clear if extended semen or raw semen is evaluated. Microbiome was evaluated in raw semen so to avoid the antibiotic effect of extender’s antibiotics (line 89). Sperm quality parameters were evaluated in a dilution in commercial extender (clarified in lines 87-88).

  1. The range of total motility (68-94%) in the stallion population is not very large. The authors report that is associated with a Peptoniphilaceae and higher motility and lower motility with Clostridiales Incertae Sedis X. Thank you for your comment. Total motility was high in all stallions. However, it was possible to distinguish two groups: average and high total motility. Since we did not collect any ejaculate with total motility < 50% we cannot completely confirm this hypothesis (lines 176-177). Sperm Morphology was not assessed which could account for the variability between stallions in terms of sperm motility parameters. Unfortunately, we did not have enough stallions with relevant morphology abnormalities and we did not evaluate morphology in a daily basis.

  1. The cut-offs used were not evidence based to indicate semen quality. No references are provided. The cut-off values were chosen according to Love [3]. Based on this reference, there is an internal law in the Department of Defence that states that only ejaculates yielding sperm progressive motility >30% are considered acceptable to send. DNA fragmentation does not have a consensual cut-off value; however, we used the published references [4,5].

  1. Recommend that this data is better presented as a descriptive study on the semen microbiome and that the fertility data is presented to demonstrate the stallion’s have normal semen parameters. We have changed the title, objective, and parts in the discussion. We have also avoided the term fertility.

Specific Comments

  1. Title: Suggestion The Semen Microbiome and Semen Parameters in Healthy Stallions. Title changed as suggested.

  1. Objective: to review the associations between the presence of bacteria and sperm quality parameters. The experimental design does not allow this to be determined. Objective changed.

  1. Line 67-68 please provide more information on what the stallions were fed. Trace minerals? Grain? Information included (lines 85-86).

  1. Line 70 and 76 Re use of the extraction, do you mean semen collection? Please clarify. Thank you for your correction. Correction done (lines 74-75).

  1. Line 72-73 In preparation for semen collection was the stallion’s penis prepared or washed in any way? This is normally a standard procedure prior to semen collection. Please describe. Penis was washed with warm water after abstinence periods of a month or more, or if there is too much visible smegma. The study was performed during breeding season, so no specific penis preparation was done before sample collection. This Was a phantom used for semen collection or was the stallion collected off of the mare? We used a phantom mare (information included in line 81).

  1. Line 75 Were the stallion’s individually housed? As the results don’t agree with other authors is this an environmental element? Yes, stallions were individually housed in boxes (included in line 74). To the best of our knowledge, environment does not have a relevant impact on seminal microflora. It is noteworthy to say that most of the studies referred to men [1,2,6,7]. However, there are still some few papers about stallions [8,9], where different housing did not affect microbiome variability.

  1. Line 77 It would be better to state that the raw semen was frozen for analysis. Corrected (line 89).

  1. Line 85 Is this a sterile tube? Yes, it was a sterile tube (included, line 98). However, the aliquot of fresh semen for microbiome analysis was immediately taken and cryopreserved after collection (line 89).

  1. Line 87 Sperm motility is influenced by extenders such as INRA 96. Was the raw semen evaluated for motility? The same is true for DNA fragmentation, was the raw semen or extended semen used for the DNA fragmentation analysis? Extended semen was used for both motility and DNA fragmentation analysis up to a final concentration of 25 x 106 sperm/ml (lines 102-103). You are right that commercial diluents improve sperm quality parameters. However, this effect is usually observed some time (one hour) after dilution [10]. In our study, quality assessment was performed immediately after collection, so we consider that results have not benefit from the diluent effect.

  1. Line 114 pooled? Describe what and how many samples were pooled? No, there were no sample pooling. Corrected (line 137).

  1. Line 135 Why was sperm morphology not evaluated? Morphology is only evaluated in the breeding center the study was performed when animals show fertility issues. Unfortunately, although it would have been a really interesting quality parameter to include, we did not have enough stallions with relevant morphological abnormalities.

  1. Line 137-141 A per cycle pregnancy rate is the most important statistic to use regarding stallion fertility. Is this information available? This should be used to divide the stallions into high and low rather than the arbitrary cut offs. How many mares were bred? Thanks for the comment, per cycle pregnancy rate is a much better value to evaluate stallion fertility. This information is not completely available, as many doses are used for mares that are not under our control. If so how many progressively motile morphologically normal sperm were the mares bred with? Regular doses contain 500x106 progressively motile normal sperm.

  1. These cut-off values should have a biologic basis. What literature supports these cut-offs if any? Cut-off values were chosen according to Love [3]. We have also used an internal law of the Department of Defence (>30% or progressive motility) in order to send seminal doses. DNA fragmentation does not have a consensual cut-off value; for this study, we considered [4,5]. It is necessary to state that we inferred the cut-off value according sperm concentration and total number of spermatozoa the volume parameter. All of these parameters are influenced by the frequency of semen collection. This would influence the category within which the stallion’s sperm parameters are placed. How do you account for this? For sure the overuse of a stallion dramatically changes their sperm quality parameters. This point was controlled by performing the study during breeding season (spring and summer), where animals are regularly submitted to collections (information included in line 78). During this season, animals are submitted to a maximum of three collections per week, with a 48 hour rest interval between collections. For our study, we used the ejaculate collected the day of semen demand.

  1. Line 145 - 146 Table 1: there are no red or green numbers on the Table. Colours added. Samples were collected from 21 stallions why are there 12 stallions in the Table? Thank you for the correction. There was a typo. The total number of stallions is 12.

  1. Line 149 Table how many stallions per group? You state you collected 21 stallions. That was a mistake. There were 12 stallions. The number of stallions per group varied in function of the quality parameter. For example, regarding concentration, there were 6 stallions in the high concentration group, and 6 in the low concentration group. However, with progressive motility, there were 4 animals in the high progressive motility group, and 8 in the low motility group. Members of each group are now coloured in Table 1.

  1. Line 164 How many stallions per group? This method of data plotting should be changed to stacked bar graphs Graph changed.

  1. Bray-Curtis dissimilarity plots? This inclusion would be of great interest, but we feel that it lies beyond the objective of this study. Our article wants to be just a first approach of a possible correlation between bacteria and sperm quality parameters in healthy stallions. We think that it would be of the utmost interest to include them if there were infertile or subfertile stallions. If you consider this plots essential for the study, we can include them.

  1. Line 172 there will be a correlation if you calculate it , significant or not, but that does not prove causality. To the best of our knowledge, there is no literature in regards to the interactions between spermatozoa and seminal flora. Current papers can only describe or associate statistically the presence/absence of bacteria to sperm quality, diseases or even fertility. Proving causality may imply finding chemical mechanisms, which is now a step further from the current state of knowledge.

  1. The discussion does not appear to cover possible environmental factors as a cause of the variability in microbiome. Factors influencing seminal microflora has been studied overall in men [1,2,6,7]. According to these studies, microflora shifts are mostly affected by diseases, sexual intercourses or medical treatments. However, little environmental factors have not proven to have a significant effect on microflora composition. Regarding the stallion, and despite the low number papers available [8,9], current evidence maintain the theory that environmental factors does not have a significant effect on microbiome variability.

References:

  1. Hou, D.; Zhou, X.; Zhong, X.; Settles, M.L.; Herring, J.; Wang, L.; Abdo, Z.; Forney, L.J.; Xu, C. Microbiota of the Seminal Fluid from Healthy and Infertile Men. Fertil. Steril. 2013, 100, 1261–1269, doi:10.1016/j.fertnstert.2013.07.1991.
  2. Weng, S.-L.; Chiu, C.-M.; Lin, F.-M.; Huang, W.-C.; Liang, C.; Yang, T.; Yang, T.-L.; Liu, C.-Y.; Wu, W.-Y.; Chang, Y.-A.; et al. Bacterial Communities in Semen from Men of Infertile Couples: Metagenomic Sequencing Reveals Relationships of Seminal Microbiota to Semen Quality. PLoS ONE 2014, 9, e110152, doi:10.1371/journal.pone.0110152.
  3. Love, C.C. Sperm Quality Assays: How Good Are They? The Horse Perspective. Anim Reprod Sci 2018, 194, 63–70, doi:doi: 10.1016/j.anireprosci.2018.04.077.
  4. Sergerie, M.; Laforest, G.; Bujan, L.; Bissonnette, F.; Bleau, G. Sperm DNA Fragmentation: Threshold Value in Male Fertility. Hum. Reprod. 2005, 20, 3446–3451, doi:10.1093/humrep/dei231.
  5. D’Occhio, M.J.; Hengstberger, K.J.; Johnston, S.D. Biology of Sperm Chromatin Structure and Relationship to Male Fertility and Embryonic Survival. Animal Reproduction Science 2007, 101, 1–17, doi:10.1016/j.anireprosci.2007.01.005.
  6. Altmäe, S.; Franasiak, J.M.; Mändar, R. The Seminal Microbiome in Health and Disease. Nat Rev Urol 2019, 16, 703–721, doi:10.1038/s41585-019-0250-y.
  7. Tomaiuolo, R.; Veneruso, I.; Cariati, F.; D’Argenio, V. Microbiota and Human Reproduction: The Case of Male Infertility. High Throughput 2020, 9, 10, doi:10.3390/ht9020010.
  8. Al-Kass, Z.; Guo, Y.; Vinnere Pettersson, O.; Niazi, A.; Morrell, J.M. Metagenomic Analysis of Bacteria in Stallion Semen. Anim Reprod Sci 2020, 221, 106568, doi:10.1016/j.anireprosci.2020.106568.
  9. Quiñones-Pérez, C.; Hidalgo, M.; Ortiz, I.; Crespo, F.; Vega-Pla, J.L. Characterization of the Seminal Bacterial Microbiome of Healthy, Fertile Stallions Using next-Generation Sequencing. Anim Reprod 2021, 18, e20200052, doi:10.1590/1984-3143-AR2020-0052.
  10. Ghallab, A.M.; Shahat, A.M.; Fadl, A.M.; Ayoub, M.M.; Moawad, A.R. Impact of Supplementation of Semen Extender with Antioxidants on the Quality of Chilled or Cryopreserved Arabian Stallion Spermatozoa. Cryobiology 2017, 79, 14–20, doi:10.1016/j.cryobiol.2017.10.001.

Round 2

Reviewer 1 Report

The authors explained all the objections raised to the manuscript in their responses. The text has been refined enough.

Author Response

Reviewer 1:

General comments:

The authors explained all the objections raised to the manuscript in their responses. The text has been refined enough.

Authors’ response: We would like to thank this reviewer for the time reviewing this paper. Their contributions have significantly improved the quality of the manuscript.

Reviewer 2 Report

The semen microbiome and semen parameters in healthy stallions.

Semen was opportunistically collected from 11 stallions using an artificial vagina. The semen was subdivided and raw semen was processed for DNA extraction and NGS. The other fraction was extended and semen was evaluated for qualtiy. Authors divide semen into quality parameters of high/low and evaluate rationships with bacterial phyla.

Authors appear to be trying to very hard to demonstrate that stallion sperm quality is associated with the sperm microbiome features rather than just reporting their findings in health stallions. The stallions they are studying are all have acceptable sperm characteristics. Dividing them arbitrarily into high and low parameters is not appropriate .

  1. Introduction the data on aerobic culture results from stallions should be included.

2. Materials and methods

Authors must demonstrate that the samples were not contaminated. Did the semen collection personnel wear gloves? if the stallion's penis was not prepared prior to collection this should be described. Any lubricants should be listed.

Stallion housing in terms of bedding should be described.

3. You indicate that CASA was performed and list the variables such as VSL, ALH and then you do not analyze these variables. Report on what you analyzed.

4. Classification of "stallions" These threshold values are arbitrary, affected by collection frequency, and are not appropriate. Report the sperm parameter values for the stallions, do not divide them. A stallion's fertility is based on the whole picture the total number of progressive morphologically normal sperm. Dividing them by parameter into high and low in inappropriate. Rather than classifying the whole ejaculate as acceptable or not or using fertility data (per cycle conception rate). These are all fertile stallions based on what is presented. 

5. Table 1 list stallion semen parameters only

6. Data analysis for table 2 is inappropriate.

7. Data should be shown for phyla present by stallion.

Ideally the core microbiome dominant groups and distributions should be shown along with the relative abundance of bacterial phyla by stallion. If you are describing the health stallion determine if there are differences between stallions. Correlate semen parameters to bacterial phyla in healthy stallions as a group.

Specific comments

Line 16 replace “conditions” with affects sperm quality

Line 22 replace “associate” with to examine associations with ….

Line 23 remove volume (It is part of concentration)

Line 26 replace “worse” with lower

Line 26 replace may be behind to “may contribute to”

Line 34 replace “associate” with to examine associations with ….

Line 34 …horses …add “and there is limited information on the microflora present in stallion ejaculates.”

Line 51 Cut “In order to measure,” start sentence with There are objective

Line 60 these articles do not show clear causality with sperm prebiotics rather associations change to some “positive effects in improving”

Line 64 change ”linking” to evaluating

Line 66 change to five (remove volume line 66)

Line 85 – 86 Stallions were collected a maximum of three times per week. Add detailed information on the preparation of the penis prior to collection.

Line 87 what was the mean or median stallion age +/- SD or quartiles Refer to Table 1

Line 88 estrus

Line 94 minerals?

Line 100 remove “their”

Line 120 remove “filled”

Line 127 was this raw semen?

Line 131 There is really no basis for these threshold values to divide the ejaculates of the stallions using these numbers. Gel free volume, by itself is not useful, it is used to calculate concentration.

Line 146 add “raw” semen

2.2.5 Statistical analysis

Your current statistical analysis is meaningless.

Determine if there are differences between stallions in the Phyla

Calculate correlations with the whole data, do not divide the stallions into groups. You have one group healthy stallions.

This data should be presented by stallion using the stacked bar graphs to show individual healthy stallion variability in the main bacterial phyla in microbiome in their ejaculates

Table 1 Do not dividing the data into high and low.  Simply report characteristics of the ejaculates.

Remove figure 2.  See above comments

Discussion

Rewrite discussion and remove all references to high and low. This classification is arbitrary and misleading.

Line 243 remove low and high quality

Discussion of the factors known to affect microflora briefly infections, medications, cancer etc.

Author Response

Reviewer 2:

General comments

The semen microbiome and semen parameters in healthy stallions.

Semen was opportunistically collected from 11 stallions using an artificial vagina. The semen was subdivided and raw semen was processed for DNA extraction and NGS. The other fraction was extended and semen was evaluated for qualtiy. Authors divide semen into quality parameters of high/low and evaluate rationships with bacterial phyla.

Authors appear to be trying to very hard to demonstrate that stallion sperm quality is associated with the sperm microbiome features rather than just reporting their findings in health stallions. The stallions they are studying are all have acceptable sperm characteristics. Dividing them arbitrarily into high and low parameters is not appropriate

Authors’ response: We thank this reviewer for their valuable comments. We have followed their recommendations reporting the results obtained in ejaculates from healthy stallions.

  1. Introduction the data on aerobic culture results from stallions should be included. Authors’ response: We have included a short sentence referring that most of reference are culture-based studies in the introduction (lines 61-63). The results of these studies have been also discussed.
  2. Materials and methods

Authors must demonstrate that the samples were not contaminated.

Authors’ response: Some information regarding the measures taken for contamination prevention is now included in lines 85-87.

Did the semen collection personnel wear gloves?

Authors’ response: Yes, they did. This information is now included (line 85).

If the stallion's penis was not prepared prior to collection this should be described. Authors’ response: This information has been added in the sample collection section (line 89).

Any lubricants should be listed.

Authors’ response: Done (line 88).

Stallion housing in terms of bedding should be described.

Authors’ response: Information included (line 90).

  1. You indicate that CASA was performed and list the variables such as VSL, ALH and then you do not analyze these variables. Report on what you analyzed.

Authors’ response:  Thank you for the correction. Since the variables were not analyzed, these kinematic parameters have been deleted from the M&M section, as they were not presented in the Results and Discussion sections.

  1. Classification of "stallions" These threshold values are arbitrary, affected by collection frequency, and are not appropriate. Report the sperm parameter values for the stallions, do not divide them. A stallion's fertility is based on the whole picture the total number of progressive morphologically normal sperm. Dividing them by parameter into high and low in inappropriate. Rather than classifying the whole ejaculate as acceptable or not or using fertility data (per cycle conception rate). These are all fertile stallions based on what is presented. 

Authors’ response: This is a very interesting comment. We have omitted the classification of stallions in the whole manuscript.

  1. Table 1 list stallion semen parameters only

Authors’ response: Done. The table has been modified as suggested

  1. Data analysis for table 2 is inappropriate.

Authors’ response: Table 2 has been substantially changed. We have now included the Bray-Curtis dissimilarity in order to evaluate microflora differences between animals.

  1. Data should be shown for phyla present by stallion.

Authors’ response: Done (Table 2).

Ideally the core microbiome dominant groups and distributions should be shown along with the relative abundance of bacterial phyla by stallion. If you are describing the health stallion determine if there are differences between stallions. Correlate semen parameters to bacterial phyla in healthy stallions as a group.

Authors’ response: Done. Figure 2 now represents relative abundance of bacterial phyla by stallion. In order to determine differences between stallions, Bray-Curtis dissimilarity was calculated (Table 2). This index is usually recommended when analysing the abundance of bacterial diversity in metagenomics.

Specific comments

Line 16 replace “conditions” with affects sperm quality

Authors’ response: Correction done.

Line 22 replace “associate” with to examine associations with ….

Authors’ response: Correction done.

Line 23 remove volume (It is part of concentration)

Authors’ response: Correction done

Line 26 replace “worse” with lower

Authors’ response: Correction done

Line 26 replace may be behind to “may contribute to”

Authors’ response: Correction done.

Line 34 replace “associate” with to examine associations with ….

Authors’ response: Correction done.

Line 34 …horses …add “and there is limited information on the microflora present in stallion ejaculates.”

Authors’ response: Correction done.

Line 51 Cut “In order to measure,” start sentence with There are objective

Authors’ response: Correction done.

Line 60 these articles do not show clear causality with sperm prebiotics rather associations change to some “positive effects in improving”

Authors’ response: Correction done.

Line 64 change ”linking” to evaluating

Authors’ response: Correction done.

Line 66 change to five (remove volume line 66)

Authors’ response: Correction done. lower

Line 85 – 86 Stallions were collected a maximum of three times per week.

Authors’ response: Correction done.

Changed.

Add detailed information on the preparation of the penis prior to collection.

Authors’ response: This is information is now included in sample collection.

Line 87 what was the mean or median stallion age +/- SD or quartiles Refer to Table 1. Authors’ response: Done (line 82).

Line 88 estrus

Authors’ response: Done.

Line 94 minerals?

Authors’ response: No minerals were added.

Line 100 remove “their”

Authors’ response: Done.

Line 120 remove “filled”

Authors’ response: Done.

Line 127 was this raw semen?

Authors’ response: No, it was diluted up to a final concentration of 25 x 106 sperm/ml in INRA 96® (information in lines 92-94).

Line 131 There is really no basis for these threshold values to divide the ejaculates of the stallions using these numbers. Gel free volume, by itself is not useful, it is used to calculate concentration.

Authors’ response: Threshold values and gel free volume deleted from the paper.

Line 146 add “raw” semen.

Authors’ response: Done.

2.2.5 Statistical analysis

Your current statistical analysis is meaningless. Determine if there are differences between stallions in the Phyla.

Authors’ response: Done. Now Table 2 represents Bray-Curtis dissimilarity between stallions.

Calculate correlations with the whole data, do not divide the stallions into groups. You have one group healthy stallions.

Authors’ response: Done.

This data should be presented by stallion using the stacked bar graphs to show individual healthy stallion variability in the main bacterial phyla in microbiome in their ejaculates.

Authors’ response: Done (Figure 2).

Table 1 Do not dividing the data into high and low.  Simply report characteristics of the ejaculates.

Authors’ response: Done.

Remove figure 2.  See above comments.

Authors’ response: Done.

Discussion

Rewrite discussion and remove all references to high and low. This classification is arbitrary and misleading. Line 243 remove low and high quality.

Authors’ response: Allusions about high and low have been removed from the discussion.

Discussion of the factors known to affect microflora briefly infections, medications, cancer etc.

Authors’ response: We have found it difficult to find specific studies regarding factors affecting seminal microflora. Most of papers about seminal composition and infections/cancer usually refer to specific genus or families, not to the general seminal flora. Our results only partially agree with literature. That makes it almost impossible to extrapolate their conclusions to our results. In the discussion, alternately, we have summarized the conclusions from other authors about families we have found in our study.